# TESA: TASK-AGNOSTIC EMBEDDING OF SCENE GRAPHS USING VISION ALIGNMENT

## ABSTRACT

Scene Graphs (SGs) provide richer information, stronger structure, and greater interpretability compared to raw images. However, existing SG embedding models are typically trained in a task-specific manner, limiting their ability to generalize across downstream applications. To address this, we introduce **T**ask-Agnostic **E**mbedding of **S**cene Graphs using Vision **A**lignment (TESA). TESA employs a Graph Neural Network trained with a cosine similarity loss to align ground-truth SG embeddings with image embeddings produced by a frozen, pretrained foundation model. This design leverages the generalization properties of foundation models and transfers them into the SG domain. We evaluate embedding quality on three datasets (VG-150, PSG, and GQA) and assess generalization on four downstream tasks: Image Retrieval, Visual Question Answering, Scene Graph Generation, and Image Classification. Our experiments show that replacing visual embeddings generated by foundation models with TESA embeddings yields comparable or even improved performance. These results demonstrate that TESA produces high-quality, task-agnostic SG embeddings that retain the structural and interpretability advantages of SGs, while achieving effectiveness on par with image-based representations.

## 1 INTRODUCTION

Recently, models capable of producing high-quality image embeddings usable across multiple domains have emerged. These Vision Foundation Models (VFMs) (Dosovitskiy et al., 2021; Kirillov et al., 2023) are trained on millions of images to produce high-quality image embeddings. Furthermore, Vision Language Models (VLMs) (Radford et al., 2021; Zhai et al., 2023) integrate language as an additional modality by aligning language and images in a common embedding space. VFMs and VLMs are valuable not only for vision-related tasks but also for various other research domains. For instance, they are frequently employed in robot learning to embed visual or textual representations without the need to train separate vision or language encoders (Reuss et al., 2024; Blank et al., 2024; Octo Model Team et al., 2024; Zitkovich et al., 2023).

In contrast, current Scene Graph (SG) embedding models are almost exclusively trained in a task-specific manner, which severely limits their generalizability. To address this, we introduce **Task-Agnostic Embedding of Scene Graphs using Vision Alignment (TESA)**, a method that provides "out-of-the-box" SG embeddings for downstream tasks without requiring retraining for each new application. SGs, displayed in Figure 3), capture a high-level abstraction of visual data, including relationships between objects, semantic labels, and visual features per object. Prior work has shown that structured image representations such as SGs can improve performance in vision-related tasks, like Image Retrieval (IR) (Johnson et al., 2015), Visual Question Answering (VQA) (Zhang et al., 2019; Hildebrandt et al., 2020) or image captioning (Yang et al., 2019; Nguyen et al., 2021). Compared to images, SGs abstract away pixel-level details, making them more robust to noise and other background changes (Johnson et al., 2015).

Achieving these advantages requires **a model capable of embedding SGs** consistently across different tasks. TESA is the first model creating SG embeddings using a Graph Neural Network (GNN) which are applicable zero-shot to different tasks. TESA is trained and evaluated on large scale SG datasets. To demonstrate the robustness of TESA, we additionally evaluate TESA on automatically generated SGs. During training, TESA aligns graph embeddings with vision embeddings from

a frozen, pretrained VFM or VLM, such as *DIstillation of Knowledge with NO labels (DINOv2)* (Oquab et al., 2024), *Contrastive Language-Image Pretraining (CLIP)* (Radford et al., 2021), or *Sigmoid Loss for Language Image Pre-Training (SigLIP)* (Zhai et al., 2023). Using a contrastive loss the graph and image embeddings are pushed towards a unified representation. The GNN design choice ensures that TESA is permutation invariant and can handle any graph size while being simple in network complexity. TESA does not aim to surpass existing foundation models but rather serves as an additional model for embedding SGs by leveraging pre-trained VFMs or VLMs.

We evaluate the SG embeddings on IR and test their task-agnostic capabilities on VQA, Scene Graph Generation (SGG), and Image Classification. For IR, we also run experiments on generated SGs to verify that TESA works when ground-truth annotations are unavailable. The goal for task-agnostic settings consists of achieving similar results when swapping between graph and image embedding. The main contribution of this paper can be summarized as follows: We introduce TESA, a novel framework for task-agnostic SG embedding. We align SG embeddings across 4 different foundation models, showcasing the flexibility of TESA.

## 2 RELATED WORK

The following sections provide an overview of representative methods for the tasks of Image Retrieval, Visual Question Answering, and Scene Graph Generation. All approaches rely on visual input in some form, enabling the exchange between image and graph embeddings as input.

### 2.1 IMAGE RETRIEVAL

The IR problem setting tries to find images to a corresponding query, where the query can be any embedding, related to the image (Datta et al., 2008; J. & R., 2022; Yan et al., 2021). Example features include color and shape (Jain & Vailaya, 1996) or textual cues (Mishra et al., 2013). Early work (Johnson et al., 2015) already showed that using SGs has advantages over other modalities to retrieve images. Follow-up methods retrieved images by leveraging already existing SG embedding methods (Schroeder & Tripathi, 2020). Current SOTA approaches introduce the idea of learning similarities between images and SGs (Yoon et al., 2021). They even introduce a new metric to evaluate IR in the context of SGG (Cong et al., 2024).

### 2.2 VISUAL QUESTION ANSWERING

VQA aims at image understanding through question answering, where answers can vary from binary Yes/No, over numbers to many word answers (Antol et al., 2015; Schwenk et al., 2022; Shao et al., 2023; Hudson & Manning, 2019). The domain of VQA includes a variety of different datasets (Wu et al., 2017). Traditional VQA approaches rely on image features to answer questions (Antol et al., 2015; Wu et al., 2017; Shih et al., 2016), whereas current SOTA models include Large Language Models (LLMs) (Shao et al., 2023; Li et al., 2023; Seenivasan et al., 2023). SGs can be used to enhance the VQA process (Zhang et al., 2019; Hildebrandt et al., 2020; Nuthalapati et al., 2021).

### 2.3 SCENE GRAPH GENERATION

SGG can be structured into two main categories: one- and two-stage approaches. Two stage approaches firstly predict bounding boxes and labels and afterwards predicates between the given objects (Zellers et al., 2018; Yang et al., 2018; Zheng et al., 2023; Yan et al., 2020; Wang et al., 2023; Tang et al., 2019). First stage approaches do all these steps at once (Cong et al., 2023; Li et al., 2022). Generating SGs requires different information given by the image: Object bounding boxes, object labels, object features, global image features. Additionally, including prior knowledge can help with SGG, where the co-occurrences of objects and predicates given by the dataset are used (Zellers et al., 2018; Wang et al., 2023).

### 2.4 SCENE GRAPH EMBEDDINGS

SG embeddings are useful for different downstream tasks. Methods train their embedding network dependent on the task at hand, making the SG embedding network less flexible. Task-dependent SG

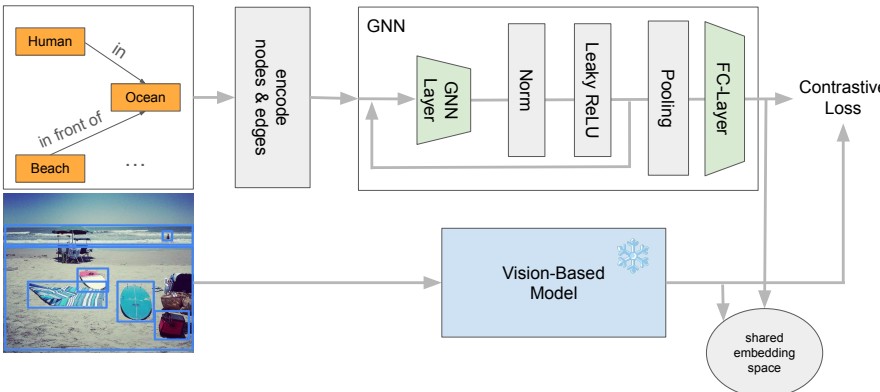

Figure 1: TESA takes all objects and relations from a given SG and produces a graph representation, where the initial node features are text CLIP embeddings. Afterward, a GNN embeds the graph, using GNN layers, pooling and a fully connected layer, resulting in one final embedding vector. For training, the ground truth embeddings are generated, using the vision-based model and compared to the graph embeddings. By optimizing the contrastive loss from Equation (1), TESA is able to embed SGs into a shared space with the image embeddings.

embeddings can be found in the context of autonomous driving, where observations are converted into a SG (Malawade et al., 2022). The model predicts collisions based on the current SG and adapts its embedding network. Other task-dependent approaches use layout generation (Schroeder et al., 2019) or image generation (Yang et al., 2022b) to produce useful SG embeddings. Another method trains graph embeddings on prior text, describing the scene (Maheshwari et al., 2021).

One approach called Scene Graph-Image Contrastive Learning Framework (SPAN) (Cong et al., 2024), proposes a related idea to TESA. SPAN is based on two Transformer (Vaswani, 2017) based networks for both image and graph embedding. **Their architecture choice** is grounded on the assumption that the locality of GNNs makes them unfit for capturing SG representations (Cong et al., 2024). Our IR evaluations on TESA do not support this claim, suggesting GNNs as a valid architecture choice for SG embeddings. Further, SPAN needs a special structural encoding to preserve the structure of the input graph, uses random shuffling to be permutation invariant, and uses a maximum of 10 nodes for a SG. In contrast, TESA avoids these limitations by design and is the only task-agnostic SG embedding model, which is actually evaluated on downstream tasks.

## 3 METHOD

As described in Section 2.4, the goal of TESA is to generate SG embeddings that can be used for downstream tasks using a GNN. The embeddings should be *task-agnostic*, implying that they are useful beyond the task-specific training objective. Therefore, TESA leverages foundation model embeddings, to transfer their generic characteristic to TESA. The task-agnostic capabilities of TESA will later be evaluated on VQA, SGG and Image Classification. The problem can therefore be stated as follows: How to generate SGs embeddings that align to equivalent vision-based embeddings in a task-agnostic context?

### 3.1 LOSS FUNCTION

The idea of creating SG embeddings with vision alignment draws inspiration from the vision-language alignment of CLIP (Radford et al., 2021). TESA tackles the challenge of learning *similarly well-performing* embeddings by contrastive learning. TESA uses the cross entropy loss,

$$\mathcal{L}_{\text{cross}}(B) = -\frac{1}{|B|} \sum_{y \in B} \log \frac{\exp(\text{sim}(\mathbf{y}_{\text{graph}}, \mathbf{y}_{\text{image}}))}{\sum_{z \in B} \exp(\text{sim}(\mathbf{y}_{\text{graph}}, \mathbf{z}_{\text{image}}))}, \tag{1}$$

which maximizes the similarity between related graph and image embedding pairs ($\mathbf{y}_{graph}$, $\mathbf{y}_{image}$) while pushing away the graph embedding from other image embeddings ($\mathbf{z}_{image}$). The whole idea of the training process is illustrated in Figure 1, including the network architecture.

## 3.2 NETWORK ARCHITECTURE AND SCENE GRAPH ENCODING

TESA is a GNN based architecture displayed in Figure 1. SGs are converted into input graphs by either one-hot or CLIP-text encoding the objects and relations as initial node features. We decided for CLIP-text encodings, as it is more general compared to one-hot encodings even though the performance is worse, as seen in Table 11. Each object and relation instance in the SG obtains a node in the input graph. Note that there can be several nodes with the same embedding, as there can be multiple objects or relations of the same type in the scene, e.g. multiple humans or "next to" relations. Each relation node is bidirectionally connected with the subject and object nodes.

## 3.3 USING TESA FOR DOWNSTREAM TASKS

Employing either image or SG embeddings for downstream tasks requires dedicated architectures and task-specific methods. For the IR task, SG embeddings are compared directly with image embeddings, and retrieval is performed by matching the most similar pairs. A similar strategy is applied to Image Classification: both the graph and image are embedded and compared against text embeddings of the candidate labels, selecting the most similar one. These tasks operate directly on TESA without the need for additional downstream models. In contrast, VQA and SGG require an actual model that is trained on the respective task. The task-agnostic setting is still given, as the image or graph embedding is just a pre-computed input for the downstream model.

Therefore, the following architecture was chosen for VQA: A Transformer decoder, which takes as input a global image embedding produced by the foundation model, along with the tokenized question. Multihead attention pooling is used to predict the final answer. The decoder outputs are routed through five separate heads, one for each question type defined in the Generalized Question Answering (GQA) validation set. The same decoder-based approach is extended to SGG. In this case, however, the model receives not only the global image embedding but also localized features of detected objects and their initial bounding box predictions. These inputs are subsequently used to predict object, subject, and predicate labels, as well as refined bounding boxes.

Further architectural details are provided in Appendix E. Both architectures are trained using only the global image embedding provided by the foundation model, which during inference can be interchanged with the TESA embedding. This allows for a simple evaluation to determine if the TESA embedding is as powerful as the image embedding. Additionally, it enables us to remove the global embedding completely, testing if the model even requires this representation for prediction.

## 4 EXPERIMENTAL SETTINGS

We conduct experiments to answer the following research questions:
**RQ1**: Is TESA able to embed SG into the same embedding space as the corresponding images?
**RQ2**: How well do SG embeddings from TESA align to corresponding text-description embeddings of the image?
**RQ3**: Are SG embeddings from TESA as powerful on unseen tasks as the corresponding image embeddings?
TESA is evaluated using four different vision-based models: Two VLMs, CLIP (Radford et al., 2021) and SigLIP (Zhai et al., 2023); one VFM, DINOv2 (Oquab et al., 2024); and a simple pre-trained ResNet50 (He et al., 2016). The ResNet50 acts as a sanity check, to confirm the expected higher performance of the foundation models. TESA is trained 12 times, separately for each combination of dataset and vision-based model. The resulting models are named accordingly: TESA$_{CLIP}$, TESA$_{DINOv2}$, TESA$_{SigLIP}$ and TESA$_{ResNet50}$, where the subtext denote the embeddings used to train TESA. During training, the train-val-test split provided by the dataset was used whenever available. Otherwise, we use 70% of the data for training, 10 % for validation and 20 % for testing. Further hyperparameters and more details can be found Appendix A. Ablation results can be found in Appendix G.

## 4.1 DATASETS

TESA is trained and evaluated on three different datasets: Visual Genome (VG-150) (Xu et al., 2017), Panoptic Scene Graphs (PSG) (Yang et al., 2022a) and Generalized Question Answering (GQA) (Hudson & Manning, 2019). VG-150 provides 88,000 data points, PSG 49,000 and GQA 85,000. All three include image data, bounding boxes and SGs. GQA (Hudson & Manning, 2019) additionally includes questions for the provided images. More information about the datasets are listed in Table 5. All three datasets incorporate the Common Objects in Context (COCO) dataset (Lin et al., 2014). This results in image overlap, but due to re-annotated object and relation labels, the SG overlap is minimal. Therefore, training and testing on all three datasets gives valuable insights.

## 4.2 TASKS AND METRICS

We evaluate TESA on four different tasks to address the given research questions. For **RQ1** (embedding performance), we measure cosine similarity between TESA-based graph embeddings and image embeddings. Additionally, the performance on IR is evaluated to validate the alignment quality. For **RQ2**, text and graph embeddings are directly compared using cosine similarity. They are visualized using $TESA_{SigLIP}$ on PSG and the overlapping COCO subset via Principal Component Analysis (PCA) (Abdi & Williams, 2010). Downstream tasks (VQA, SGG, and Image Classification) assess task-agnostic performance of SG embeddings (**RQ3**). Prior work highlights the benefits of SGs for VQA (Zhang et al., 2019; Hildebrandt et al., 2020), and we extend this analysis to a broader set of downstream tasks.

**Image Retrieval** retrieves relevant images from a database given a query, which may be text, visuals, or in this work, a SG (Datta et al., 2008). Evaluation uses R-Precision@K (Cong et al., 2024), comparing K SGs with K images, with higher K increasing difficulty. Values of 10, 50, and 100 are used. Retrieval is performed by encoding SGs with TESA and matching them to image embeddings from the corresponding vision models.

**Visual Question Answering** is a scene understanding task where models answer questions about images (Wu et al., 2017). Evaluation is based on answer accuracy across categories. GQA defines 12 categories, but only Query, Logical, Verify, Choose, and Compare are used here, as others are absent in the validation split.

**Scene Graph Generation** generates SGs directly from images, evaluated under three tasks: Predicate Classification, Scene Graph Classification (SGCls), and Scene Graph Detection (SGDet). TESA focuses on SGCls and SGDet, where SGs must be produced from image and bounding boxes or image alone. Evaluation uses Recall@K and MeanRecall@K (Chen et al., 2019). Recall@K measures correct triplets within the top K predictions, while MeanRecall@K averages results across predicates to reduce bias toward frequent ones. Standard K values (20, 50, 100) are also used in TESA.

**Image Classification** requires assigning an image to a class. TESA follows CLIP (Radford et al., 2021), comparing text and image embeddings, so only CLIP and SigLIP are suitable for evaluation. Since the PSG dataset lacks classification labels, COCO images with multi-class labels are used. To simplify, only the largest bounding box label per image is considered. The task is run on the PSG train and test splits overlapping with COCO, covering 33,500 train and 10,000 test images.

## 4.3 BASELINES

The vision-based models used for training also function as baselines. Therefore, the graph-based TESA and the image-based embeddings are compared with each other, as well as other baselines, if possible. If both embedding variants perform equally for any given task, meaning they could just be swapped without loss of performance, then this confirms that TESA is a task-agnostic SG embedding model. The image-based models are denoted with $Image_{ResNet50}$, $Image_{SigLIP}$, $Image_{DINOv2}$ and $Image_{CLIP}$. In the VQA and SGG setting additional baselines are introduced denoted as *Zero-Vector*. These baselines input a vector filled with zeros instead of image or graph embedding to check if the model actually relies on these inputs or can solve the task without global image/graph information. Other baseline approaches are explained in Appendix F.

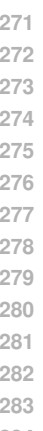

Figure 2: The PCA evaluation of the TESA SG embeddings using TESA$_{SigLIP}$, displayed as red crosses, compared to the image embeddings in blue and the text embeddings in black. The 20 data points are randomly taken from the PSG test-split with captions from COCO.

Table 1: **Image Retrieval**: Evaluation on the VG-150, PSG and GQA test-split. R-P@K stands for R-Precision@K and is measured in percent.

| Dataset | Method | R-P@10 | R-P@50 | R-P@100 |
|---|---|---|---|---|
| VG-150 | CLIP$_{Text}$ (Cong et al., 2024) | 79.80 | 57.80 | 47.20 |
| | SPAN (Cong et al., 2024) | 86.70 | 67.90 | **58.00** |
| | TESA$_{ResNet50}$ | 75.94 | 45.75 | 33.20 |
| | TESA$_{DINOv2}$ | 88.32 | 61.96 | 47.37 |
| | TESA$_{CLIP}$ | 88.53 | 64.24 | 49.85 |
| | TESA$_{SigLIP}$ | **91.36** | **70.65** | 57.36 |
| PSG | TESA$_{ResNet50}$ | 79.14 | 49.10 | 36.63 |
| | TESA$_{DINOv2}$ | 91.30 | 69.66 | 55.05 |
| | TESA$_{CLIP}$ | 91.59 | 70.55 | 56.96 |
| | TESA$_{SigLIP}$ | **93.79** | **77.33** | **64.46** |
| GQA | TESA$_{ResNet50}$ | 77.95 | 47.82 | 35.14 |
| | TESA$_{DINOv2}$ | 90.29 | 66.31 | 51.11 |
| | TESA$_{CLIP}$ | 90.94 | 69.14 | 56.08 |
| | TESA$_{SigLIP}$ | **93.04** | **75.58** | **63.49** |

## 5 EXPERIMENTAL RESULTS

Quantitative cosine similarity results for all three datasets and four vision-based models are reported in Table 6. TESA embeddings are most similar to image embeddings on PSG and GQA, with the highest overall similarity achieved by TESA$_{ResNet50}$ (0.53–0.56). While this suggests better downstream performance, the assumption does not hold: embedding closeness does not necessarily yield high-quality embeddings, as performance also depends on the underlying model. This motivates IR as a follow-up evaluation to better assess embedding quality.

Qualitative embedding results (Figure 2) illustrate 20 image–graph–text triplets from COCO. Graph embeddings (red crosses) align closely with corresponding image embeddings (blue dots). They spread evenly without collapsing, while also clustering near text embeddings (black dots) despite not being trained on them. Quantitative embedding results, across 10,045 triplets, show that text–graph cosine similarity (0.29) exceeds image–text cosine similarity (0.12). Image–graph cosine similarity (0.42) is highest overall, reflecting the training objective. These findings show that TESA embeds SGs close to both image and text spaces, thereby addressing **RQ2** and partly **RQ1**.

Table 2: **Visual Question Answering**: Evaluation on the GQA questions of type query, compare, choose, logical, and verify on the validation-split. Accuracy is in percent.

| Method on GQA | Query | Comp. | Choose | Log. | Verify |
|---|---|---|---|---|---|
| GQA$_{Vis}$ (Hudson & Manning, 2019) | 1.55 | 36.34 | 0.85 | 47.18 | 47.02 |
| GQA$_{VisLang}$ (Hudson & Manning, 2019) | 31.80 | 56.62 | 61.40 | 62.05 | 67.00 |
| MAC (Hudson & Manning, 2018) | 38.91 | 60.04 | 70.59 | 69.99 | 75.45 |
| PrVQA Shevchenko et al. (2020) | 39.31 | 62.21 | **71.12** | 71.14 | **76.17** |
| TMN-Tree Yamada et al. (2024) | **48.0** | - | 47.2 | 64.0 | 56.0 |
| Image$_{DINOv2}$ | 40.63 | 60.86 | 67.96 | 75.90 | 70.25 |
| TESA$_{DINOv2}$ | 39.94 | 60.64 | 65.92 | 75.50 | 69.91 |
| Zero-Vector$_{DINOv2}$ | 18.57 | 59.97 | 57.96 | 65.12 | 60.19 |
| Image$_{CLIP}$ | 40.16 | **63.74** | 68.56 | **77.34** | 71.51 |
| TESA$_{CLIP}$ | 37.81 | 63.16 | 66.24 | 77.28 | 70.73 |
| Zero-Vector$_{CLIP}$ | 15.34 | 62.29 | 57.67 | 65.05 | 59.09 |
| Image$_{SigLIP}$ | 43.85 | 62.87 | 69.07 | 76.87 | 71.85 |
| TESA$_{SigLIP}$ | 41.13 | 61.80 | 66.33 | 76.26 | 70.17 |
| Zero-Vector$_{SigLIP}$ | 16.47 | 59.97 | 57.83 | 64.27 | 58.41 |

## 5.1 IMAGE RETRIEVAL

Image Retrieval is evaluated on all three datasets using ground truth SGs displayed in Table 1. Qualitative results for IR on the PSG dataset can be seen in Figure 4. For quantitative results on the VG-150 dataset, TESA is compared to two baseline methods. TESA$_{CLIP}$ uses the SG embeddings directly, compared to the baseline CLIP$_{Text}$, which produces text embeddings based on the given SGs. TESA$_{CLIP}$ has a higher performance on all three categories compared to CLIP$_{Text}$, seen in Table 1. These findings suggest, that embedding SGs directly instead of converting them to text is more efficient for IR. The performance of TESA$_{ResNet50}$ is lower compared to any of the foundation models, even though its test cosine similarity was the highest, as shown in Table 6. TESA$_{SigLIP}$ achieves the highest result for @10 and @50 and is almost equal to SPAN for @100.

These trends can also be observed for the PSG and GQA datasets. The overall performance of all four models increases for the GQA dataset and again for the PSG dataset. The results for IR over all three datasets conclude in two observations. First, high test cosine-similarity values do not always correspond to high IR scores, as shown by TESA$_{ResNet50}$ and TESA$_{DINOv2}$. Therefore ResNet50 is not further evaluated in downstream tasks. Second, the on average higher R-P@K values for the PSG dataset could correspond to higher quality annotations per image. The PSG and VG-150 dataset have the same amount of predicates per image, but different IR results. This observation suggests that a higher quality of triplets per image could lead to better information that is embedded, therefore making alignment with vision embeddings easier. These findings, together with the cosine-similarity evaluation, give a concrete answer for **RQ1**. TESA model SG embeddings are closely embedded to their corresponding image embeddings. Further, using those embeddings for IR yields results that are equal or better compared to given baselines.

Results for IR on VG-150 and PSG with generated SGs is presented in Table 7. For the PSG dataset performance drops for all evaluated models, but for the VG-150 dataset it actually increases. A more in depth analysis of these results can be found in Appendix D.2.

## 5.2 VISUAL QUESTION ANSWERING

VQA is used as an unseen downstream task for TESA, as no embedding model was trained or optimized specifically for this setting. Table 2 reports results across six baselines alongside models based on image embeddings, TESA embeddings, and a zero-vector input. Our models were all trained on the image embeddings as a global token that can be swapped during inference for the graph or zero-vector embedding. The model trained with Image$_{CLIP}$ embeddings achieves the highest accuracy in two of the five question categories. Across all categories, replacing image embeddings with SG embeddings leads to only a minor decrease in performance between 0.1 and 2.5 percent. In contrast, using zero vectors results in a substantial performance drop across all categories.

Table 3: **Scene Graph Generation**: Evaluation on the VG-150 and PSG test-split. All evaluation results use graph-constraints and the sub-category of Scene Graph Classification (SGCls). PSG baselines do not offer results for SGCls.

| Dataset | Method | R@20 | R@50 | R@100 | mR@20 | mR@50 | mR@100 |
|---|---|---|---|---|---|---|---|
| VG-150 | KERN (Chen et al., 2019) | - | **36.7** | **37.4** | - | 9.4 | 10.0 |
| | RelTR (Cong et al., 2023) | **29.0** | 36.6 | - | 7.7 | 11.4 | - |
| | SGG via PK (Wang et al., 2023) | - | - | - | **18.52** | **22.23** | **23.44** |
| | Image$_{DINOv2}$ | 25.52 | 26.54 | 27.29 | 13.77 | 14.29 | 14.73 |
| | TESA$_{DINOv2}$ | 24.65 | 25.79 | 26.41 | 12.39 | 12.91 | 13.17 |
| | Zero-Vector$_{DINOv2}$ | 14.83 | 15.08 | 15.14 | 7.44 | 7.57 | 7.60 |
| | Image$_{CLIP}$ | 25.97 | 27.30 | 28.05 | 14.40 | 15.14 | 15.57 |
| | TESA$_{CLIP}$ | 27.55 | 29.25 | 30.25 | 15.39 | 16.36 | 16.94 |
| | Zero-Vector$_{CLIP}$ | 17.57 | 17.93 | 18.03 | 8.93 | 9.08 | 9.12 |
| | Image$_{SigLIP}$ | 25.82 | 26.73 | 27.37 | 14.21 | 14.67 | 14.99 |
| | TESA$_{SigLIP}$ | 27.87 | 29.16 | 29.98 | 15.69 | 16.30 | 16.66 |
| | Zero-Vector$_{SigLIP}$ | 16.03 | 16.27 | 16.35 | 7.86 | 7.97 | 8.00 |
| PSG | Image$_{DINOv2}$ | 40.51 | 44.42 | 45.59 | 37.03 | 40.76 | 41.52 |
| | TESA$_{DINOv2}$ | 41.03 | 43.36 | 44.12 | 36.32 | 38.23 | 38.68 |
| | Zero-Vector$_{DINOv2}$ | 26.90 | 27.49 | 27.74 | 24.38 | 24.71 | 24.85 |
| | Image$_{CLIP}$ | 42.69 | 44.72 | 45.24 | 39.59 | 41.37 | 41.83 |
| | TESA$_{CLIP}$ | 44.19 | 46.47 | 47.04 | 40.59 | 42.53 | 43.07 |
| | Zero-Vector$_{CLIP}$ | 32.61 | 33.85 | 33.91 | 28.13 | 28.98 | 29.0 |
| | Image$_{SigLIP}$ | 42.89 | 45.02 | 45.58 | 38.94 | 40.76 | 41.15 |
| | TESA$_{SigLIP}$ | **44.65** | **46.86** | **47.37** | **41.22** | **42.97** | **43.36** |
| | Zero-Vector$_{SigLIP}$ | 24.54 | 24.83 | 24.88 | 21.56 | 21.72 | 21.74 |

These results yield three key insights. First, the trained models rely on the global token for predictions. Second, embeddings produced by TESA generalize to previously unseen tasks such as VQA. Third, TESA enables embedding SGs into the same representation space as images, while maintaining competitive downstream performance on VQA. Together, these findings provide partial evidence toward answering **RQ3**.

## 5.3 SCENE GRAPH GENERATION

SGG constitutes the second downstream task for testing TESA embeddings. Downstream models are trained on the VG-150 and PSG datasets using image embeddings as global representation tokens, which can be replaced at inference time with either TESA or zero-vector embeddings. The latter serves again as a control to assess whether the global embedding contributes meaningfully to the SGG process. Although it seems paradox to provide the model with the embedding of the SG it should predict, the model is trained solely on image embeddings, making TESA embeddings a fair aligned alternative rather than task-specific supervision. For both datasets we evaluate SGCls, displayed in Table 3, and SGDet, displayed in Table 8. All models are compared to the same baselines, expect for SGCls where no PSG results are reported.

For SGCls none of our models can outperform the baselines, but replacing image embeddings with graph embeddings consistently improves performance for models initialized with VLM-based representations. For SGDet (Table 8) on the VG-150 dataset, TESA embeddings achieve the best results, outperforming all baselines. In the SGDet case graph embeddings improve performance on the VG-150 dataset, but not on the PSG dataset. In all cases, zero-vector embeddings lead to substantial performance degradation, confirming the importance of meaningful global embeddings.

These findings highlight two key insights. First, TESA embeddings can improve SGG performance when using VLMs for alignment. Second, results with DINOv2 embeddings suggest a closer alignment between text and graphs when TESA is trained on VLM embeddings rather than purely visual ones. Together, these results provide further evidence toward answering **RQ3**.

Table 4: **Image Classification**: Evaluation on the PSG train and test split, where images are overlapping with the COCO dataset. Accuracy is in percent.

| Split | Image$_{CLIP}$ | TESA$_{CLIP}$ | Image$_{SigLIP}$ | TESA$_{SigLIP}$ |
|-------|------|------|------|------|
| Train | 37.9 | **42.2** | 40.32 | **42.58** |
| Test | 37.94 | **42.4** | 40.14 | **42.36** |

## 5.4 IMAGE CLASSIFICATION

Image Classification is evaluated in a zero-shot manner where first, the image or graph is embedded and second, the text label is embedded. The label is not directly embedded, but using a descriptive sentence: 'In this photo you can see $< object >$'. Therefore, only the CLIP and SigLIP based model could be used to evaluate this task. The results are displayed in Table 4, which clearly show higher performance when using TESA embeddings instead of image embeddings. These insights finalize a specific answer for **RQ3**: Graph embeddings are as powerful and in some task settings even more powerful compared to the corresponding image embeddings.

## 6 LIMITATIONS AND FUTURE WORK

The performance of TESA is constrained by the used vision-based model, as shown by the ResNet50 results. If the vision model performance is in general bad, the TESA performance will also be in a similar range. The same applies to the used SG datasets, where quality and quantity determine the generalization performance. TESA is a **first step towards** a foundation model for SGs, but limited by the SG dataset sizes. CLIP was trained on over 400 million image-text pairs, whereas TESA is trained on only 50,000 to 80,000 image-graph pairs. Extending evaluations to additional downstream tasks would further clarify the broader utility of SG-based embeddings. In general, future improvements could include better foundation models, as TESA benefits from higher quality models. Generated SGs could increase the size and quality of datasets, which could increase overall performance and generalizability. Additionally, training downstream models on image and graph embeddings would also be interesting to see if it improves overall performance. Finally, hybrid solutions in which TESA functions as one component within a broader framework may open up new opportunities for leveraging SGs in multimodal learning.

## 7 CONCLUSION

We proposed **T**ask-Agnostic **E**mbedding of **S**cene Graphs using Vision **A**lignment (TESA). TESA is a GNN based method, trained on four vision-based models and on three different datasets. Quantitative and qualitative embedding results confirm the success of our training approach, where TESA was evaluated on Image Retrieval (IR) and three downstream tasks. In IR it achieves SOTA performance for the R-Precision@K metric. The VQA task results using graph embeddings are on par with equivalent methods using image embeddings reported in other papers. The SGG results are better compared to established baselines and improve further when using graph embeddings instead of image embeddings. The Image Classification results are unexpected, where TESA embeddings lead to better performance compared to the image embeddings. Our results indicate that through the aligning process, SG specific information is not lost and can be used in downstream applications. Overall, the concept of TESA is simple and flexible, making it a useful asset for any task that requires high-quality SG embeddings. Sticking with TESA could bridge the gap between foundation models and Scene Graphs.

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

# APPENDIX

## A TESA HYPERPARAMETERS

The hyperparameters used for TESA across all three datasets and both tasks are as follows:

- Number of epochs: 400
- Learning rate: 0.001
- Learning rate scheduler: Yes
- Warm up: Yes
- Number of GNN layers: 2
- Layer type: GIN
- Dropout: 0.1
- Pooling type: Max
- Hidden dimension: 512

The learning rate scheduler consists of a linear scheduler for the warm-up face, which is 20 epochs long, and a step scheduler for the other 380 epochs. The pooling type describes how all nodes merged into one embedding at the final step.

Table 5: All three datasets used in this paper are based on the COCO (Lin et al., 2014) dataset alone or also include the Flickr 30K (Young et al., 2014) dataset. PSG and VG-150 include images and SGs, and GQA additionally provides visual questions. PPI describes counts of predicates per image.

| Dataset | #Img | #Obj | #Rel | #PPI | Source |
|---------|------|------|------|------|--------|
| PSG | 49K | 133 | 56 | 5.7 | Annotate COCO |
| VG-150 | 88K | 150 | 50 | 5.7 | COCO & Flickr |
| GQA | 85K | 1703 | 310 | 50.6 | Re-annotate VG |

## B   DATASET INFORMATION

The used datasets have COCO (Lin et al., 2014) as their underlying image dataset and VG-150 additionally uses Flickr (Young et al., 2014). SGs, as displayed in Figure 3, consist of different object-predicate-subject triplets. The PSG and VG-150 dataset have 5.7 predicates per image, whereas GQA has over 50.

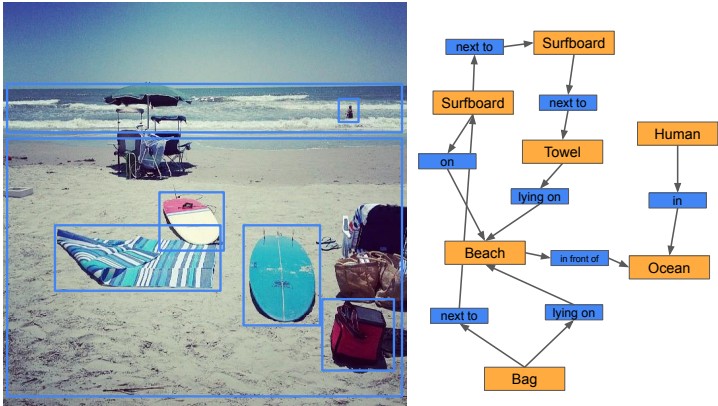

Figure 3: Example image from COCO (Lin et al., 2014) with a custom created SG. Note that this SG is not complete and only resembles one possible graph for this image. The detected objects in the image are represented by the orange graph nodes. The possible relations between objects are displayed as blue nodes.

## C   USAGE OF LARGE LANGUAGE MODELS

For related work LLMs were used after finding most of the related papers to make sure nothing was missed. Additionally, LLMs were used to refine some already written paragraphs in the paper. It also helped with writing code for evaluating the downstream tasks.

## D   ADDITIONAL RESULTS

We report cosine similarities, additional results for IR on generated graphs and the SGDet for the VG-150 and PSG dataset.

### D.1   COSINE SIMILARITIES

The cosine similarity values per model are reported in Table 6. The ResNet50 model has the highest similarity, even though it has the lowest performance on IR. TESA$_{\text{SigLIP}}$ and TESA$_{\text{CLIP}}$ have lower similarity values compared to TESA$_{\text{DINOv2}}$. This result could be due to the additional text alignment in the embedding space of the VLMs, which makes it harder to additionally align SGs.

Table 6: **Cosine Similarity**: Evaluation on all three datasets with regard to the test-split. The reported values are the cosine similarity, where 1 means their angle is exactly the same and -1 their angle is directly opposite.

| Method | VG-150 | PSG | GQA |
|---|---|---|---|
| TESA$_{ResNet50}$ | **0.53** | **0.56** | **0.55** |
| TESA$_{SigLIP}$ | 0.40 | 0.42 | 0.41 |
| TESA$_{DINOv2}$ | 0.50 | 0.54 | 0.51 |
| TESA$_{CLIP}$ | 0.41 | 0.44 | 0.42 |

Table 7: **Image Retrieval**: Evaluation on the VG-150, PSG and GQA test-split using ground truth (gt) and generated (gen) SGs from our own SGG models. R-P@K stands for R-Precision@K and is measured in percent.

| Dataset | Method | R-P@10 | R-P@50 | R-P@100 | #PPI |
|---|---|---|---|---|---|
| VG-150 | TESA$_{DINOv2}$ (gt) | 88.32 | 61.96 | 47.37 | 5.7 |
| | TESA$_{DINOv2}$ (gen) | 89.38 | 63.86 | 48.98 | 32.8 |
| | TESA$_{CLIP}$ (gt) | 88.53 | 64.24 | 49.85 | 5.7 |
| | TESA$_{CLIP}$ (gen) | 89.76 | 65.90 | 51.98 | 39.7 |
| | TESA$_{SigLIP}$ (gt) | 91.36 | 70.65 | 57.36 | 5.7 |
| | TESA$_{SigLIP}$ (gen) | **92.27** | **71.41** | **58.08** | 40.6 |
| PSG | TESA$_{DINOv2}$ (gt) | 91.30 | 69.66 | 55.05 | 5.7 |
| | TESA$_{DINOv2}$ (gen) | 90.86 | 67.68 | 52.48 | 28.7 |
| | TESA$_{CLIP}$ (gt) | 91.59 | 70.55 | 56.96 | 5.7 |
| | TESA$_{CLIP}$ (gen) | 89.84 | 65.55 | 50.45 | 31.4 |
| | TESA$_{SigLIP}$ (gt) | **93.79** | **77.33** | **64.46** | 5.7 |
| | TESA$_{SigLIP}$ (gen) | 92.69 | 72.42 | 59.32 | 27.4 |

## D.2 IMAGE RETRIEVAL ON GENERATED SCENE GRAPHS

In Table 7, we observe a consistent divergence in the effect of using ground truth versus generated SGs across datasets. While both VG-150 and PSG have a similar number of ground truth predicates per image (PPI) of 5.7, the impact of generatedSGs differs substantially. On VG-150, generated graphs yield notable improvements in retrieval performance across all backbones. In contrast, on PSG, generated graphs consistently degrade performance.

This discrepancy can be attributed to differences in annotation quality. VG-150 is known to suffer from noisy and incomplete scene graph annotations, often omitting valid relations. The generated graphs, although noisier and significantly denser with 30-40 PPI, provide additional relational evidence that compensates for these gaps and benefits retrieval. Conversely, PSG provides more reliable and comprehensive annotations. In this case, the additional relations introduced by generated graphs primarily act as noise, overwhelming the high-quality ground truth signal and reducing performance.

Overall, these results indicate that the utility of generated scene graphs for image retrieval is strongly dependent on the underlying dataset quality: they are beneficial when annotations are incomplete but detrimental when the ground truth is already rich and accurate. Further steps would include no only training TESA on ground truth graphs but also on generated ones. This could improve performance and generalizability.

## D.3 SCENE GRAPH DETECTION

Table 8 displays the SGG results for SGDet, where only the image is given as input to the model. Our trained models outperform almost all other baselines. For the VG-150 dataset using TESA embeddings instead of image embeddings increases results. This is not the case for the PSG dataset, where results decrease when using TESA embeddings. For both datasets and all models, using zero-vector embeddings decreases performance significantly.

Table 8: **Scene Graph Generation**: Evaluation on the VG-150 and PSG test-split. All evaluation results use graph-constraints and the sub-category of Scene Graph Detection (in some papers it is called Scene Graph Generation instead of SGDet).

| Dataset | Method | R@20 | R@50 | R@100 | mR@20 | mR@50 | mR@100 |
|---|---|---|---|---|---|---|---|
| VG-150 | KERN (Chen et al., 2019) | - | 27.1 | 29.8 | - | 6.4 | 7.3 |
| | RelTR (Cong et al., 2023) | 21.2 | 27.5 | - | 6.8 | 10.8 | - |
| | SGG via PK (Wang et al., 2023) | - | - | - | 12.44 | 15.99 | 19.21 |
| | Image$_{CLIP}$ | 30.17 | 41.90 | 48.03 | 17.19 | 26.82 | 33.70 |
| | TESA$_{CLIP}$ | 32.09 | 44.42 | 50.97 | 18.10 | 28.70 | 35.93 |
| | Zero-Vector$_{CLIP}$ | 4.70 | 7.04 | 8.42 | 3.25 | 4.94 | 5.87 |
| | Image$_{SigLIP}$ | 31.60 | 44.24 | 50.90 | 18.42 | 29.28 | 37.02 |
| | TESA$_{SigLIP}$ | **32.76** | **45.92** | **52.35** | **18.92** | **29.90** | **37.65** |
| | Zero-Vector$_{SigLIP}$ | 1.68 | 2.86 | 3.51 | 0.79 | 1.35 | 1.78 |
| PSG | PSGTR (Yang et al., 2022a) | 28.2 | 32.1 | 35.3 | 15.4 | 20.3 | 21.5 |
| | HiLo (Zhou et al., 2023) | **40.6** | 48.7 | 51.4 | 29.7 | 37.6 | 40.9 |
| | PairNet (Wang et al., 2024) | 33.3 | 39.3 | 42.4 | 25.4 | 28.2 | 29.7 |
| | Image$_{CLIP}$ | 38.52 | **51.55** | **59.19** | **38.40** | **51.73** | **59.01** |
| | TESA$_{CLIP}$ | 36.74 | 50.12 | 58.07 | 35.92 | 50.28 | 57.80 |
| | Zero-Vector$_{CLIP}$ | 6.31 | 9.86 | 11.70 | 5.17 | 8.23 | 10.21 |
| | Image$_{SigLIP}$ | 36.95 | 49.90 | 57.71 | 36.90 | 50.42 | 57.88 |
| | TESA$_{SigLIP}$ | 36.54 | 48.97 | 56.10 | 34.46 | 47.56 | 54.23 |
| | Zero-Vector$_{SigLIP}$ | 3.18 | 4.41 | 5.28 | 3.47 | 4.43 | 5.05 |

# E  VQA AND SGG ARCHITECTURE

Both the VQA and SGG models adopt a Transformer decoder architecture, with task-specific input tokens and output heads.

For VQA, the input sequence consists of a classification (`[CLS]`) token, a global image embedding (produced by CLIP, SigLIP, DINOv2, or ResNet and interchangeable with either a zero-vector or a scene graph embedding at inference), and the tokenized question. The decoder processes this sequence, and five task-specific MLP heads are used to predict answers across the five GQA question categories.

For SGG, the input extends the DETR framework by including query tokens for the subject, object, and their corresponding bounding boxes. These queries are combined with DETR's detection tokens and a global image embedding, again swappable during inference with either a zero-vector or scene graph embedding. The decoder output is passed to five MLP heads predicting the subject label, object label, predicate label, and refined bounding boxes for subject and object.

In both settings, training is conducted using only the vision-based global embeddings, while inference is used to test the effect of replacing image embeddings with scene graph embeddings.

# F  BASELINES

Baselines for IR using the VG-150 dataset are SPAN (Cong et al., 2024) and CLIP text embeddings based on SGs, denoted as CLIP$_{Text}$ (Cong et al., 2024). CLIP$_{Text}$ transforms SGs into a textual description, which is embedded by CLIP and compared to CLIP image embeddings to retrieve images (Cong et al., 2024). There are no baselines for the PSG dataset and the GQA dataset in the context of IR.

VQA baselines for GQA are taken from the GQA paper (Hudson & Manning, 2019). A Convolutional Neural Network (CNN) and a Long Short-Term Memory (LSTM) network are used to encode vision and language. The CNN-only method is denoted as GQA$_{Vis}$ and CNN plus LSTM as GQA$_{VisLang}$. Another baseline is MAC (Hudson & Manning, 2018), which uses attention networks. Additional baselines include PrVQA (Shevchenko et al., 2020), which uses prior knowledge and TMN-Tree (Yamada et al., 2024), which uses the concept of modules on Transformers (Vaswani, 2017) to answer visual questions.

Table 9: Values for different types of GNN layers all based on attention on the IR task with Precision-R@K. The used dataset was VG-150 and the TESA$_{\text{SigLIP}}$ model. Using GIN achieves the highest results, but only slightly.

| Metric | GAT | GATv2 | TransConv | GCN | GIN |
|---|---|---|---|---|---|
| R-P@10 | 90.04 | 89.59 | 90.26 | 90.96 | **91.36** |
| R-P@50 | 67.48 | 66.62 | 67.43 | 69.56 | **70.65** |
| R-P@100 | 53.50 | 52.73 | 53.77 | 56.23 | **57.36** |

Table 10: The GNN performs better than just pooling the clip-encodings of the nodes, even if embeddings are refined using an MLP. The used dataset was VG-150 and the TESA$_{\text{CLIP}}$ model.

| @K value | Mean | Max | MLP+Mean | MLP+Max | GIN+Max |
|---|---|---|---|---|---|
| R-P@10 | 36.95 | 23.39 | 86.79 | 87.25 | **88.53** |
| R-P@50 | 14.49 | 6.56 | 61.24 | 62.58 | **64.24** |
| R-P@100 | 9.44 | 3.74 | 47.00 | 48.47 | **49.85** |

SGG baselines for the VG-150 dataset include KERN (Chen et al., 2019), which uses GNNs to generate SGs, RelTR (Cong et al., 2023), which uses Transformers (Vaswani, 2017) for generation and a method that includes prior knowledge in a specific embedding (Wang et al., 2023). SGG baselines for the PSG dataset include PSGTR (Yang et al., 2022a), also a Transformer-based method for graph generation, HiLo (Zhou et al., 2023), which utilizes high and low frequencies of predicate occurrences and PairNet (Wang et al., 2024), which learns and filters sparse pair-wise relationships.

## G  ABLATIONS

Ablations focus on the usage of different GNN layers and the importance of learnable SG embeddings. In Table 9 results for 5 different GNN layers are displayed. Attention-based layers perform 3-4 percent worse for IR @100, compared to the convolutional-based Graph Convolutional Network (GCN) or Graph Isomorphism Network (GIN) layers. In general, the differences are small and demonstrate the strength of Graph Neural Network in general for embedding SGs. This observation further challenges the claim of SPAN (Cong et al., 2024) that GNNs are not able to produce high-quality SG embeddings. Therefore, adapting and training a Transformer from scratch is not necessary to achieve high IR scores on different SG datasets.

The initial node features of TESA are created using CLIP text embeddings. Therefore, aligning SG embeddings with image embeddings could be biased through their initial node embeddings, especially when using CLIP as image encoder. Ablations on the network layout displayed in Table 10 disprove this claim. Relying solely on max- or mean-pooling results in poor IR R-P@K performance. Adding in a learnable Multilayer Perceptron (MLP) after the pooling step increases performance by 50 to 63 percent. Additionally, including one GNN layer increases the performance on IR using SG embeddings. This observation confirms the importance of a learnable algorithm and displays the small but measurable performance increase using GNN layers.

The success of a model for the ablations is measured for the IR task for all three K values and on the PSG dataset using the TESA$_{\text{SigLIP}}$. Different models keep the same parameters described in Appendix A and only change one hyperparameter at a time. Ablations on the hidden dimensions revealed significant difference when dropping to 32, with highest performance for 1024, as seen in Table 13. Experiments are still conducted using 512 dimensions to reduce model complexity and training/inference time. Increasing the number of layers leads to over-smoothing of the graph information, which results in worse performance, as seen in Table 14.

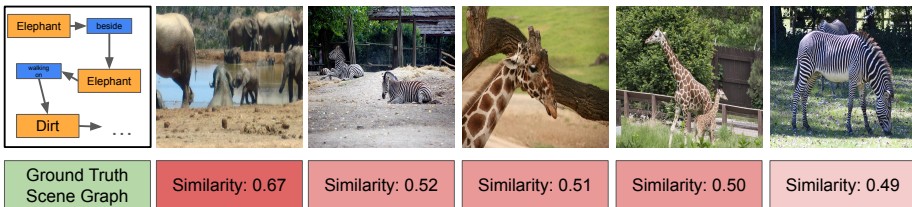

Figure 4: Qualitative evaluation of Image Retrieval using the Scene Graph of image 000000166338 from the PSG test-split. The displayed SG only shows a small portion of the complete graph, which includes many more elephants and other relations. The highest similarity is reported for the correct image. The following images are sorted descending by their similarity, given the SG. An observed underlying concept connecting all these images could be wildlife. We suspect that this characteristic is directly adopted from the SigLIP model used for training the TESA model that was used for this retrieval task.

Table 11: Comparison between initial features being one-hot encoded and text-embeddings using CLIP. Both versions use SigLIP and PSG.

| Initial Embeddings | R-P@10 | R-P@50 | R-P@100 |
|---|---|---|---|
| One-Hot | 92.30 | 70.96 | 57.76 |
| CLIP | 91.79 | 69.59 | 55.86 |

Table 12: GQA dataset question statistics. Most questions come from the Query category, also including the most possible answers.

| Question Type | # Questions | # Answers |
|---|---|---|
| Compare | 4138 | 46 |
| Choose | 16155 | 512 |
| Verify | 27331 | 2 |
| Logical | 15991 | 2 |
| Query | 67933 | 1362 |

Table 13: Values for different hidden dimensions on the IR task with Precision-R@K. The used dataset was VG-150 and the TESA$_{\text{SigLIP}}$ model. There are no significant differences, suggesting any hidden dimension value is viable between 32 and 1024.

| @K value | 16 Dim. | 64 Dim. | 256 Dim. | 512 Dim. (TESA) | 1024 Dim. |
|---|---|---|---|---|---|
| 10 | 54.90 | 83.80 | 90.01 | 91.36 | 91.87 |
| 50 | 19.93 | 53.31 | 67.21 | 70.65 | 72.40 |
| 100 | 11.44 | 38.51 | 53.02 | 57.36 | 59.72 |

Table 14: Values for different number of GNN GAT layers on the IR task with Precision-R@K. The used dataset was VG-150 and the TESA$_{\text{SigLIP}}$ model. Increasing the layers lowers the precision for all @K values.

| @K value | pooling + MLP | 1 Layer + MLP | 2 Layer + MLP (TESA) | 3 Layer + MLP |
|---|---|---|---|---|
| 10 | 90.65 | 91.27 | 91.36 | 90.95 |
| 50 | 69.64 | 70.48 | 70.65 | 69.77 |
| 100 | 56.81 | 57.32 | 57.36 | 56.41 |

