# OpenReview forum: "TESA: Task-Agnostic Embedding of Scene Graphs using Vision Alignment"
_ICLR.cc/2026/Conference — ICLR 2026 Conference Withdrawn Submission_

### Official Review · Reviewer_kgTE · 2025-11-01

**Soundness:** 2
**Presentation:** 2
**Contribution:** 2
**Rating:** 2
**Confidence:** 4

**Summary:**

This paper proposes TESA (Task-Agnostic Embedding of Scene Graphs using Vision Alignment), a framework that aims to learn task-agnostic representations of scene graphs (SGs) by aligning them with visual embeddings from pretrained vision foundation models such as CLIP, SigLIP, DINOv2, and ResNet50.
TESA uses a graph neural network (GNN) trained with a cosine similarity loss to map SGs into the same embedding space as images.
The authors evaluate their model on three datasets (VG-150, PSG, and GQA) and four downstream tasks: image retrieval, visual question answering (VQA), scene graph generation (SGG), and image classification.
Results show that TESA’s embeddings perform comparably, or even better in some cases, than image embeddings.

While the direction is promising, the paper remains rather preliminary. The methodology and writing are somewhat rough, and the evidence for the “task-agnostic” claim is limited. Overall, the paper feels exploratory rather than complete, and would need stronger validation, better organization, and clearer positioning to meet ICLR’s acceptance bar.

Overall, the paper explores a relevant and timely idea—transferring foundation model knowledge to scene graphs—but the current submission is not yet mature. Its methodology is simplistic, the evidence for task-agnostic generalization is weak, and reproducibility is poor due to missing implementation details. While the direction has promise, the work feels exploratory and incomplete. I recommend rejection at this stage.

**Strengths:**

1. Aligning scene graph embeddings with pretrained vision encoders is an appealing idea that connects structured and unstructured visual understanding.
2. The paper includes multiple datasets and tasks (IR, VQA, SGG, classification), showing a reasonable effort to assess generality.
3. The idea of transferring visual generalization via alignment is easy to understand and implement in principle.

**Weaknesses:**

1. Paper structure and presentation are messy. For example, Several key tables (e.g., Table 6) appear in both the main text and appendix, and core results are scattered across sections. Much of the appendix contains content that should be part of the main paper. The presentation lacks polish and coherence, reducing readability.

2. Writing quality is a bit below standard. The English is understandable but simplistic and occasionally awkward. Logical transitions are weak, and certain explanations feel repetitive or colloquial. The paper would benefit from a careful language edit.

3. The “task-agnostic” claim is not convincingly demonstrated. Despite the title, most experiments involve re-training or fine-tuning task-specific models using TESA embeddings. This setup tests whether SG embeddings can replace image embeddings, not whether they generalize across unseen tasks.

4. Research questions (RQs) don’t align with the paper’s stated goal. Among the three RQs, only RQ3 (downstream task transfer) directly relates to the notion of task-agnostic embeddings. The others are essentially similarity checks. The RQ structure feels more like post-hoc organization than a coherent research framework.

5. Comparison of fine-tuned performance is a bit away from task-agnostic SG embedding. If downstream fine-tuning is required, it should be compared fairly against fine-tuned image models. Otherwise, it remains unclear whether TESA genuinely improves or merely distills existing visual features. Methodologically, TESA is essentially CLIP-style contrastive alignment applied to graphs. The contribution is more in applying an existing paradigm than introducing a new modeling principle.

6. Reproducibility is weak. The paper provides no code, model checkpoints, or implementation details beyond a brief hyperparameter description. This makes it practically impossible for others to verify or extend the results. The claim of simplicity and reproducibility therefore does not hold.

**Questions:**

1. Were downstream models fine-tuned when using TESA embeddings? If so, were the corresponding image-based baselines also fine-tuned?
2. How much of TESA’s performance gain comes from the choice of vision encoder rather than the alignment mechanism?
3. Can TESA handle zero-shot transfer to unseen downstream tasks, or does it require retraining for each?
4. Do you plan to release code, pretrained weights, or data preprocessing scripts to ensure reproducibility?
5.Could multi-anchor alignment (using several VFMs jointly) improve embedding robustness and generalization?

---

### Official Review · Reviewer_MLNg · 2025-11-01

**Soundness:** 1
**Presentation:** 3
**Contribution:** 2
**Rating:** 2
**Confidence:** 4

**Summary:**

In TESA the authors propose an aligning scene graphs (SGs) in the same latent space as "foundational" self-supervised ViTs. Scene graphs provide inherent structural interpretive benefits such as explicit logic for VQA tasks via explicit object relations and compositional reasoning. This allows for transfer from structural graph representations to embedding spaces for existing vision models, and concomitantly, fine grained semantic details and visual cues can be transferred to existing scene graph representations---effectively providing a bridge between symbolic representations and more data driven semantic representations. The authors propose an intuitive CLIP-based approach, replacing text with scene graph embeddings while maintaining CLIP-text features for node initialisation.

However, TESA fundamentally depends on access to high-quality annotated SGs per image. These are non-trivial annotations to obtain in practice, and prior work has shown large gaps between idealized (GT) and predicted SGs, alongside risks of data leakage in evaluation protocols that condition on oracle graphs (as their own SGG setup implicitly does; they acknowledge this as “paradoxical”). These concerns echo the problem formulation in SelfGraphVQA [1], which explicitly targets leakage-free, robust learning from predicted/noisy SGs. Crucially, all downstream evaluations consume embeddings computed from ground-truth scene graphs. This constitutes oracle conditioning: unless predicted SGs are used at inference, results measure alignment under privileged supervision rather than deployable performance. Importantly, the reported scene graph generation (SGG) results do not correspond to genuine generation from images: the model is evaluated using embeddings derived from the ground-truth graphs of those same images, meaning that the input already contains the target information. This setup therefore measures only representation alignment, not generation performance, and the conclusions about SGG should be interpreted accordingly.

In summary, while TESA represents a simple, intuitive idea, the reliance on idealised SGs leads to data leakage. This has been reported as a central methodological issue for SG methods in previous works, and remains a problem for TESA since high quality annotated SGs are not typically available for images. This reviewer would even argue that pure text annotations are easier to come by than ground truth SGs. The practical utility of TESA is therefore wholly reliant on additional high level labels in deployment, which are difficult to come by.

[1] [Souza et al. 2023 - SelfGraphVQA: A Self-Supervised Graph Neural Network for Scene-based Question Answering](https://arxiv.org/abs/2310.01842)

**Strengths:**

1. The idea is clear, intuitive, and relatively simple.
2. TESA is modular and relatively inexpensive. A small GNN aligns SGs to frozen vision model spaces via a CLIP-style contrastive loss (Eq. 1); which makes the approach easy to integrate.
3. The experimental results cover a lot of ground (IR, VQA, SGG, zero-shot classification) with generally competitive numbers; retrieval competitive with SPAN despite using a simple GNN. However, the issue remains that using idealized ground truth SGs is appending very strong labels in tasks, resulting in inherent data leakage.

**Weaknesses:**

1. Downstream experiments use idealized embeddings from ground-truth scene graphs at inference, not predicted graphs; this is oracle conditioning across tasks (IR, VQA, SGG, classification), hence the reported results measure alignment under privileged supervision rather than deployable performance.
2. The model uses only label-initialized nodes and their connections, without geometry, bounding boxes, or attributes. Because there are no tests such as edge-shuffling or collapsing relation labels, it remains unclear whether the reported gains come from actual relational structure or simply from the semantic content of the labels embedded in CLIP space.
3. Results with generated SGs vary by dataset (improve on VG-150, degrade on PSG), indicating some inherent sensitivity to annotation regime and likely label-coverage effects rather than actual structural reasoning.
4. The “surprising” zero-shot classification gains are under-analyzed and plausibly reflect language-space regularization rather than actual use of graph topology. It is somewhat unclear to this reviewer what role the actual SGs play in TESA versus a sort of bag-of-labels (BoL) concept alignment.

**Questions:**

1. Can the authors provide results for downstream tasks (IR, VQA, SGG, classification) using predicted SGs at inference? What would the effect be of training TESA on non-idealized predicted SGs, with performance reported as a function of predicted SG noise?
2. What is the drop when edges are randomly rewired while preserving node-label multisets (edge-shuffle / degree-preserving rewires)?
3. How much do results degrade when all predicate labels are collapsed to a single token (e.g. a more general “related-to” edge label), keeping connectivity fixed?
4. How close does a bag-of-labels (BoL) baseline (no edges; set encoder over object labels with the same contrastive loss) get to TESA?
5. What is the effect of embedding image data to replace CLIP word embeddings for nodes?

---

### Official Review · Reviewer_DCVj · 2025-11-01

**Soundness:** 3
**Presentation:** 1
**Contribution:** 1
**Rating:** 2
**Confidence:** 4

**Summary:**

The paper identifies the problem that scene graph embeddings are trained in a task specific manner and that this might become a bottleneck when users want to use scene graph embeddings in a downstream task.

They propose the solution of using a graph neural network trained with cosine similarity to align the scene graph embedding with the embedding generated for the image by a foundation model.

**Strengths:**

The paper is easy to read and the idea is simple (this is a good thing).

**Weaknesses:**

The paper simply proposes to train a graph neural network to align with the embeddings generated  by a foundation model - I am not from this particular subfield of ML - but I struggle to see the novelty of this work. This paper might be seen more as a nice application report that shows how graph neural networks can be trained in conjunction with a foundation model.

The paper lacks some crucial details:

1. In equation 1 it is not clear what is $B$ (one can guess it but it is not defined)
2. Similar consideration as above holds for $y_{graph}$ and $y_{image}$
3. Still in equation (1) the authors clearly write that they use cosine similarity but they do not specify it in Section 3.1
4. The authors delegate the explanation of the architecture to Figure 1. However, in Figure 1 many details are missing including which type of pooling has been used. This is quite important because on the ground of the pooling function used the network can do different things (for example if you use sum pooling then the network can count how many nodes are there in the network, while if you use max it cannot)

In the experimental analysis:
1. For the image retrieval task the authors mainly only compared different versions of TESA
2. For the visual question answering task the authors compared with other methods, which though meant that TESA never managed to outperform the other models

**Questions:**

See Weaknesses

---

### Note · Authors · 2025-11-13

**Comment:**

We would like to express our gratitude to the Area Chair and all the reviewers for their time and effort in providing such insightful and constructive feedback on our work.

After careful consideration of the comments, we have concluded that the paper requires a substantial re-evaluation and revision that goes beyond the scope of a typical rebuttal or minor update.

We will be incorporating the valuable feedback we received as we rethink our approach and prepare this work for a more suitable venue in the future.

**Withdrawal Confirmation:**

I have read and agree with the venue's withdrawal policy on behalf of myself and my co-authors.